# Anomalous electronic transport in high-mobility Corbino rings

Sujatha Vijayakrishnan[1], F. Poitevin[1], Oulin Yu[1], Z. Berkson-Korenberg[1], M. Petrescu[1], M. P. Lilly[2], T. Szkopek[3], Kartiek Agarwal[1], K. W. West[4], L. N. Pfeiffer[4] & G. Gervais[1] ✉

We report low-temperature electronic transport measurements performed in two multi-terminal Corbino samples formed in GaAs/Al-GaAs two-dimensional electron gases (2DEG) with both ultra-high electron mobility ($\gtrsim 20 \times 10^6$ cm$^2$/ Vs) and with distinct electron density of 1.7 and $3.6 \times 10^{11}$ cm$^{-2}$. In both Corbino samples, a non-monotonic behavior is observed in the temperature dependence of the resistance below 1 K. Surprisingly, a sharp decrease in resistance is observed with increasing temperature in the sample with lower electron density, whereas an opposite behavior is observed in the sample with higher density. To investigate further, transport measurements were performed in large van der Pauw samples having identical heterostructures, and as expected they exhibit resistivity that is monotonic with temperature. Finally, we discuss the results in terms of various lengthscales leading to ballistic and hydrodynamic electronic transport, as well as a possible Gurzhi effect.

Over the last two decades, great progress has been achieved in increasing the electron mobility in two-dimensional electron gases formed in MBE-grown materials such as GaA/AlGaAs and alternatively in exfoliated graphene. Spectacularly, the electron mobility in GaAs/AlGaAs 2DEGs has recently been reported to reach $57 \times 10^6$ cm$^2$(Vs)$^{-1}$[1] and, in the absence of phonons at low temperatures, this results in large impurity-dominated mean free path that can exceed 350 µm. These high-mobility 2DEGs are notoriously well described by Fermi liquid theory at low temperatures, but what is perhaps less obvious is that counter-intuitive phenomena can arise due to an interplay between hydrodynamic transport and confinement. As an example, Gurzhi noted in 1963[2,3] that if a Fermi liquid is confined in a narrow constriction of characteristic size $d$, within some restrictive conditions the resistance of the metal could decrease with increasing temperature. More recently, studying the scattering lengths in 2D semiconductors with moderately high electron mobility, Ahn and Das Sarma[4] proposed that Gurzhi's prediction could occur even in bulk GaAs 2DEGs with sufficiently short electron-electron scattering lengths and low disorder.

Motivated by these works, we have fabricated two identical multi-terminal Corbino rings in GaAs/AlGaAs 2DEGs with electron mobility exceeding $20 \times 10^6$ cm$^2$(Vs)$^{-1}$, and with two different electron density leading to distinctive electron–electron and electron–impurity scattering lengths. Four-point conductance (resistance) measurements were performed in these Corbino with a transport channel defined by a 40 µm annular ring probing only the bulk of the sample, i.e., with no edge. These measurements are compared with similar measurements performed in millimeter-scale van der Pauw (VdP) samples that have the same heterostructure. We note that the intrinsic resistivity (conductivity) in the VdP and Corbino samples differs from the measured resistance (conductance) solely by a geometric factor. Therefore, both will be used interchangeably in the text below. Astonishingly, the temperature dependence of both multi-terminal Corbino shows an anomalous temperature dependence whereby in one case a sharp decrease in resistance is observed at temperatures below 1 K with increasing temperature, and an opposite behavior is observed in the other Corbino sample. In the van der Pauw (VdP) samples with identical heterostructure to the

[1]Department of Physics, McGill University, Montréal, Québec, Canada. [2]Center for Integrated Nanotechnologies, Sandia National Laboratories, Albuquerque, NM, USA. [3]Department of Electrical and Computer Engineering, McGill University, Montréal, Québec, Canada. [4]Department of Electrical Engineering, Princeton University, Princeton, NJ, USA. ✉e-mail: gervais@physics.mcgill.ca

multi-terminal Corbino, as expected a monotonic behavior of resistivity with temperature was observed.

# Results

## Corbino and van der Pauw geometry

Typical measurements performed in large 2DEG samples usually utilizes either samples prepared in van der Pauw or Hall bar geometry. In the VdP geometry, the electrical contacts are located in each corner and midpoint along the perimeter of a square wafer of size $3 \times 3$ mm in our case. Four-point measurements are then performed by applying a fixed current (voltage) and measuring voltage (current) using a combination of four contacts. Even though non-patterning helps to preserve the pristine state of the ultra-high quality 2DEG, transport measurements performed in either VdP or Hall bar unavoidably contain both the bulk and surface (or more precisely the edge) effects. In the case where only the bulk contribution to the resistivity (or conductivity) is wanted, one can have recourse to the Corbino geometry where the sample contacts are prepared in a circular/ring geometry, with the active region of the 2DEG sample defined by an annulus. In this case, the current is applied concentrically from the outer to inner radii and the electronic transport measurements solely probe the bulk of the sample. Most studies performed so far in the Corbino geometry have focused on samples with an inner contact and a single outer ring and these are inherently two-point contact measurements where unfortunately the contact resistance with the 2DEG is not eliminated. However, multi-terminal Corbino samples possessing one inner contact and three outer rings can be fabricated for four-point measurements, see Fig. 1a. While the Corbino geometry does not have any edge nor radial dimension, however, it does have a channel length which is defined as the distance between the inner diameter of the third ring ($V_+$ probe) and the outer diameter of the second ring ($V_-$ probe). In our case, this channel length is $L_{Corbino} = 40$ μm for both Corbino samples.

## Electron density and $r_s$ parameter

The dimensionless electron-electron interaction parameter $r_s = (\pi n a_B^2)^{-\frac{1}{2}}$, where $n$ is the electron density of either set of samples, and $a_B = 103$ Å is the effective Bohr magneton radius, are 0.92 and 1.32 for the 301 and 302 wafers, respectively, and they are shown in Fig. 1b. Given its significantly higher $r_s$ value and ultra-high electron mobility, the 2DEG formed in 302 samples (when compared to 301) are expected to inherently have enhanced electron–electron interaction, and conversely significantly distinct electron–electron scattering lengths versus temperature.

## Four-point Corbino conductance measurements

Electronic transport measurements were carried on with the multi-terminal Corbino samples. In this geometry, it is usual to measure the conductance and for this reason, we first show the measured conductance versus temperatures in both samples. We performed these measurements with two distinct measurement circuits, labeled setup A (20 nA excitation current) and setup B (83 nA excitation current), see supplementary material (SM). These results are shown in Fig. 2a for CBM301 and Fig. 2b for CBM302 and the data obtained with each circuit are in excellent agreement with one another. The temperature dependence of conductance measured in CBM301 shows a decrease in conductance at temperatures above ~400 mK, which at first sight may not be surprising given there have been reports of increasing electron mobility well below 1 K in 2DEGs of moderately high mobility[5]. Surprisingly, a completely opposite trend is observed for CBM302 where an anomalous increase in conductance (decrease in resistance) with increasing temperature is observed at temperatures above ~500 mK. The resulting fractional change in conductance ($\Delta G/G$) is ~40%

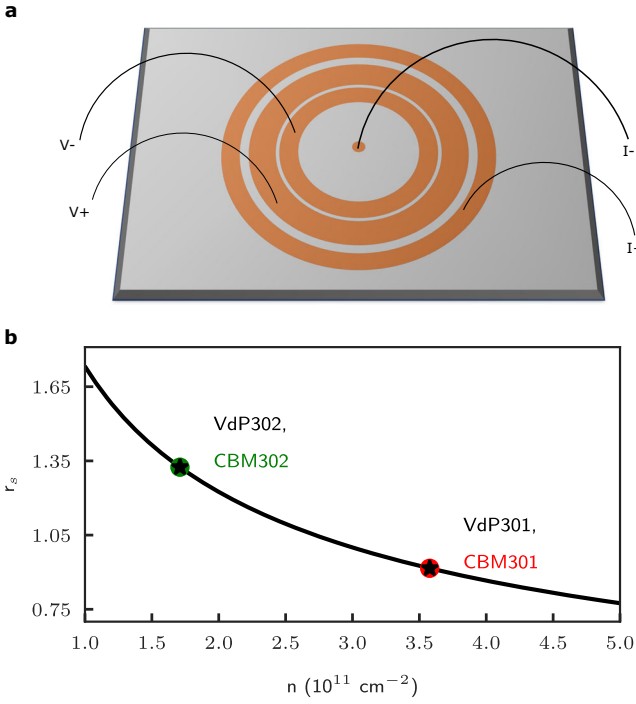

**Fig. 1 | Schematics of the multi-terminal Corbino and electron-electron interaction parameter. a** Four-terminal Corbino design with gold rings depicting the contact geometry for samples CBM301 and CBM302. The contact ring sizes are identical for both samples. **b** Electron–electron interaction parameter $r_s$ versus electron density (see main text). The $r_s$ parameter values for CBM301 (and VdP301) denoted as red dot (star) and CBM302 (and VdP302) denoted as green dot (star) are 0.92 and 1.32, respectively.

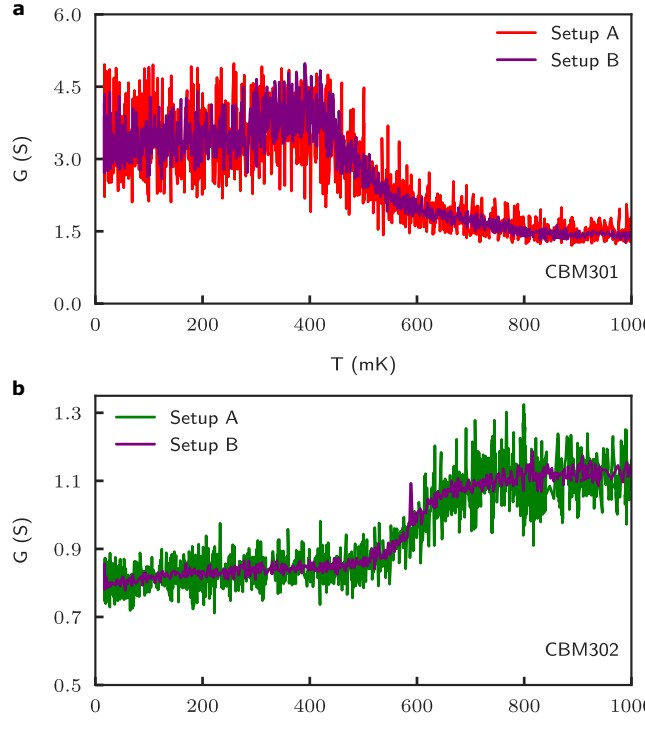

**Fig. 2 | Temperature dependence of the Corbino conductance.** The four-point conductance $G$ measured with two distinct measurement setups labeled A and B (see main text and SM), is shown *versus* temperature for CBM301 **a** and CBM302 **b**. An increase in conductance (decrease in resistance) onsetting near 500 mK is observed in CBM302 (panel **b**) with increasing temperature, whereas an opposite behavior at ~400 mK is observed for CBM301 (panel **a**). Note that in the next figures, the resistance $R = G^{-1}$ rather than the conductance will be shown.

between ~500 and 700 mK, and this is different from the anomalous transport behavior observed in CBM301 in spite of both samples having an identical geometry and similar high electron mobility.

## Probe symmetry and reciprocity theorem

To further validate our observation, we have performed conventional and unconventional transport measurements in both Corbino samples, see Fig. 3. Note that in this figure, and in the remainder of the text we opted to report and discuss the resistance $R = G^{-1}$ versus temperature rather than the conductance. In the unconventional configuration, the current ($I_+$, $I_-$) and voltage ($V_+$, $V_-$) probes are interchanged and this results in the Corbino geometry for the current probes, somewhat counterintuitively, to be connected to intermediate rings, and the voltage probes to the most outer and inner contacts, see Fig. 3a. The reciprocity theorem of electromagnetism which traces its roots to Maxwell's equations (and has been connected to Onsager's relations by Casimir[6]) states that both configurations should yield the same measurement. This holds true for passive circuits that are composed of linear media and for which time-reversal symmetry is not broken. Our reciprocity measurements are shown in Fig. 3b, c, with the data shown in magenta taken in the unconventional configuration. Except for the

higher noise floor, the measurements clearly satisfy the reciprocity theorem, adding strength to the observation of anomalous transport in the multi-terminal Corbino samples.

## Comparison of Corbino and van der Pauw measurements

van der Pauw measurements were performed to determine the bulk resistivity of each parent heterostructure. Figure 4a, and b shows the resistance measured in both Corbino and VdP samples up to 10 K temperature on a semi-log scale. The resistance of VdP301 shows a very slight decrease up to 2 K which is then followed by the expected monotonic increase due to scattering with phonons. This behavior is drastically distinct from CBM301 which exhibits a sudden increase in resistance at ~400 mK, followed by a saturation over a wide range of temperatures, up to ~10 K. In the case of VdP302, the increase in resistance due to phonons onsets near 1 K, and as expected is followed by a monotonic increase with temperature, see Fig. 4b. This greatly contrasts with the sudden decrease in resistance observed in CBM302 near ~500 mK, followed by a saturation up to a temperature of ~3 K where a monotonic increase with temperature is observed. We also note that in all cases the resistance is nearly constant at temperatures below ~0.5 K, as expected in very high-mobility GaAs/AlGaAs 2DEGs, hence ruling out the anomalous behavior in electronic transport observed being caused by a percolation process (metal-insulator transition). Recent work also found that under some conditions NiGeAu used to create the electrical contacts could be superconducting with a transition temperature of around ~700 mK and with a critical field of ~0.15 T[7,8]. In order to investigate if a supercurrent and/or a proximity effect could have played a role in our transport measurements, two-terminal and four-terminal differential resistance

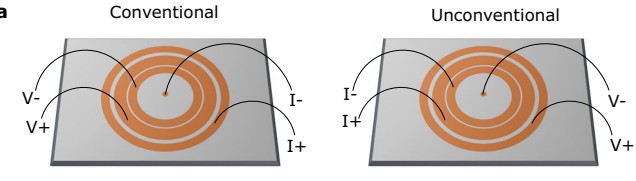

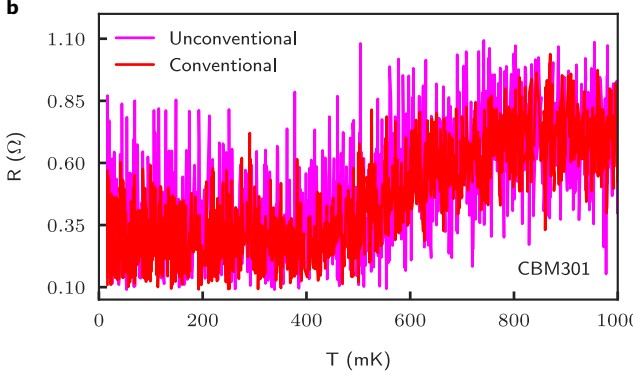

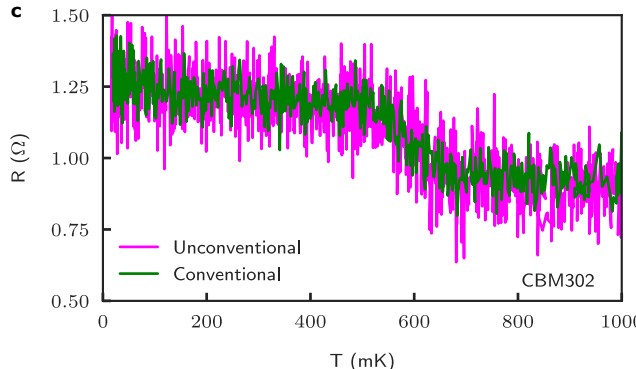

**Fig. 3 | Conventional and unconventional resistance measurement.**
**a** Schematics of the current–voltage probe terminal used for conventional (left) and unconventional (right) measurements. In the unconventional case, the current ($I_+$, $I_-$) and voltage ($V_+$, $V_-$) leads are interchanged, with the current leads connected to the inner Corbino rings. The resistance measured *versus* temperature for CBM301 (panel **b**) and CBM302 (panel **c**) is shown for both cases, with the data in magenta denoting the unconventional configuration. The reciprocity relations are clearly validated in both Corbino samples.

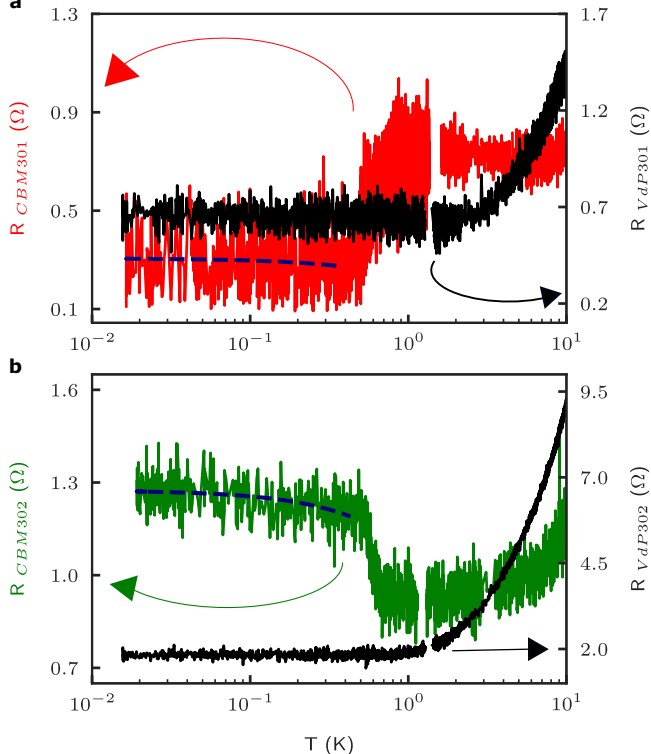

**Fig. 4 | Temperature dependence of Corbino and van der Pauw resistance (VdP).** The temperature dependence of the measured Corbino resistance is compared with the resistance measured in VdP cut from wafers with the exact same heterostructure. Panel **a** shows resistance measured for CBM301 (red) and VdP301 (black). Panel **b** shows the resistance measured for CBM302 (green), and VdP302 (black). A guide-to-the-eye blue dash line shows the moving average of the resistance at low temperatures for CBM301 and CBM302.

measurements under a DC current bias were performed at base temperature of the dilution refrigerator ($T \sim 20$ mK) and the $I - V$ obtained were found to be linear (Ohmic). In addition, we have performed magneto-transport measurements in a perpendicular field near the expected critical field 0.15 T (also at base temperature) and found no evidence of superconductivity in either sample, see SM. Finally, additional electronic transport data up to 100 K in the Corbino and up to 10 K in the VdP samples are provided in the SM that illustrates the distinctive behavior between the VdP and CBM samples at temperatures below a few Kelvins.

## Discussion

### Transport lengthscales

As discussed in Ref. 4, the relevant lengthscales that are impacting the bulk resistivity of a high-mobility 2DEG are: (A) The momentum-conserving mean free path due to electron–electron interaction, $l_{ee}$; (B) the momentum-relaxing mean free path due to electron scattering with both impurities and phonons, $l_e$; and (C) the width of the annular region forming the transport channel which is $L_{Corbino} = 40$ μm in our case. When phonons contribute negligibly to electron scattering, which has been shown to be the case below 1 K temperature[9], the momentum-relaxing mean free path $l_e$ can be calculated from the mobility and electron density, $l_e = 5.22 \times \mu\sqrt{n}$. Owing to the very high-mobility of the 2DEG used here, this length scale is exceedingly larger than the channel transport length, yielding ~271 μm in CBM301 and ~138 μm in CBM302.

Within Fermi liquid theory, Ahn and Das Sarma[4] calculated from first principles the momentum-conserving mean free path $l_{ee}$ for 2DEGs with similar density and with an electron mobility of $1 \times 10^6$ cm$^2$(V s)$^{-1}$. In particular, they found that for a 2DEG with a density $1.5 \times 10^{11}$ cm$^{-2}$ (similar to CBM302) the momentum-conserving mean free path $l_{ee}$ falls below 40 μm at temperatures above ~550 mK. At a higher electron density of $2.5 \times 10^{11}$ cm$^{-2}$ (close to CBM301), this occurs at a temperature of ~850 mK and so the condition $l_{ee} < L_{Corbino} \ll l_e$ can occur within a finite temperature interval for both CBM301 and CBM302 with a negligible phonon contribution to resistivity, i.e., until $l_e$ is reduced due to phonon scattering and becomes comparable to either $l_{ee}$ or $L_{Corbino}$.

### Knudsen parameter, ballistic transport, and hydrodynamics

In fluids, when a liquid or a gas flow through an aperture, distinctive flow regime can be classified by a dimensionless parameter known as the Knudsen number $K_n = \lambda/D$. Here, $\lambda$ is the mean free path of the fluid's constituent and $D$ is the diameter of the aperture or fluid channel. As a function of $K_n$, a classical fluid can transit from (effusive) single-particle transport described by the kinetic theory of statistical mechanics at a high Knudsen number ($K_n \gg 1$), to a continuous hydrodynamic flow governed by the Navier–Stokes at a low Knudsen number ($K_n \ll 1$), see Ref. 10 for a simple experimental demonstration in the case of a gas flowing through a small aperture.

In electronic systems such as two-dimensional electron gases, both transport regimes are in principle possible although the realization of hydrodynamic flow has proven to be challenging to observe experimentally. In their study of transport-relevant lengthscales in high-mobility 2DEGs, Ahn and Das Sarma[4] have defined a dimensionless Knudsen parameter, $\zeta \equiv l_{ee}/l_e$, which also marks the crossover from ballistic (or effusive) flow when $\zeta \gg 1$, to the hydrodynamic regime in a continuum when $\zeta \ll 1$. Interestingly, this $\zeta$ parameter falls below 0.5 at temperatures above 500 mK for both CBM301 and CBM302, hinting strongly at the occurrence of hydrodynamic transport in a regime where phonons play very little, or no role at all. We note that the sharp drop in resistance at ~500 mK observed in CBM302 occurs when $l_{ee} < L_{Corbino}$ and $\zeta \ll 1$, suggesting a hydrodynamic flow caused by increased electron-electron interaction. To our knowledge, this sharp drop (increase) observed in CBM302 (CBM301) in resistivity with

increasing temperature below 1 K has never been observed before in any high-mobility 2DEGs. Moreover, this is surprising given the transport channel length in the Corbino being 40 μm long, and for which a priori one would assume to be well within the bulk regime of 2D electronic flow.

### Gurzhi effect?

The prediction by Gurzhi that a Fermi liquid metal can have a decreasing resistance with increasing temperature due to hydrodynamic flow has been notoriously difficult to realize in 3D metals, with the end result that very little progress has been made on the topic for decades. Only over the last few years, there has been an experimental report of hydrodynamic behavior in 3D materials[11,12]. In 2D semiconductors, materials have long been made with astonishingly high-electron mobility and hence extremely long mean-free path. In spite of this, reports of hydrodynamic flow have been scarce, with the first claim made roughly thirty years ago (and only one for many years) in GaAs quantum wires[13]. More recent works have focused on bilayers[14], strongly-correlated 2D hole systems[15], non-local measurements[16,17] including an extensive study of the conductivity in GaAs/AlGaAs channel with smooth sidewalls and perfect slip boundary condition[18], graphene[19–22], graphene Corbino rings[23] as well as the theoretical interest of hydrodynamic flow in Corbino geometries[24–27]. But to the best of our knowledge, there has been no report to date of direct observation of Gurzhi's prediction of a super ballistic-to-hydrodynamic electron flow when phonons play very little, or no role at all.

The Gurzhi effect occurs when the electron flow is neither diffused by impurity scattering nor by single-particle transport events but is rather the result of the formation of a continuum and collective transport properties described by hydrodynamics. In principle, this effect can be observed in the resistivity of a clean Fermi liquid metal up to a temperature where the increasing electron-phonon interaction becomes important and leads to non-negligible scattering sources. Rather stunningly, the resistance observed below ~1 K in CBM302 shows a great similarity with the resistance curve predicted and plotted by Gurzhi in his original work[2]. This possibility is further supported by the $\zeta$ value calculated which would indicate the electron flow to be in the hydrodynamic regime. This being said, in the case of CBM301 whose higher electron density leads to a smaller $r_s$ parameter and hence with decreased electron interaction when compared to CBM302, the sudden increase in resistance around 400 mK with temperature cannot be explained solely by arguments based on hydrodynamic flow, nor by any obvious electron-phonon scattering mechanisms.

A recent study conducted on spatial mapping of local electron density fluctuation in a high-mobility GaAs/AlGaAs 2DEG by way of scanning photoluminescence[28] reported electron density variations up to 100 μm with a spot size of 40 μm. These fluctuations are likely to generate local electron mobility fluctuations, and we hypothesize that it could perhaps play a role in a Corbino measurement scheme because there is no edge and the concentric sample can be viewed as a very large number of conductors wired in parallel. This being said, whether a higher conductance path due to a local spatial fluctuation in the electron density (or mobility) would occur, and lead to the anomalous electronic transport observed in both CBM301 and CBM302 remains an open question. This will be the subject of future works.

To summarize, we have performed four-terminal electronic transport measurements in two very-high mobility Corbino 2DEG rings with distinct electron density and identical annular channel length of 40 μm. In both cases, anomalous transport was observed in the temperature dependence of the resistance at temperatures below 1 K where phonons are expected to play a negligible or no role at all. The discovery of a sharp decrease in resistance with increasing temperature in the lower density sample is stunning, and even more so since

the estimated Knudsen parameter hints at a Gurzhi effect and a crossover from super-ballistic to hydrodynamic flow. Nevertheless, this sharp decrease in resistance contrasts with the trend observed at a similar temperature in the higher electron density sample, and for which we have no clear explanation as to its origin. At temperatures above 10 K, we have shown that in both cases the expected monotonic temperature dependence of the resistance is recovered and is similar in trend to that measured in larger millimeter size VdP samples. While the exact mechanism leading to the anomalous electronic transport observed here remains an open question, this work demonstrates that a 40 μm channel length is not bulk in high mobility 2DEGs since both the momentum-conserving and momentum-relaxing mean free path values are either larger, or equal to the channel length in the 20 mK to ∼1 K temperature range. Looking forward, confirmation of hydrodynamic flow in GaAs/AlGaAs could allow us to study the remarkable properties of the anti-symmetric part of a viscosity tensor, known as the Hall viscosity[29,30], a dissipationless viscosity existing even at zero temperature that has no classical equivalent whatsoever.

## Methods

### Experimental design and procedure

The measurements were performed on GaAs/AlGaAs symmetrically-doped heterostructures with a quantum well width of 30 nm (CBM301, VdP301) and 40 nm (CBM302, VdP302). A sketch of the heterostructure that includes the main growth parameters such as the setback distances for the dopants and the capping layer thickness is provided in the SM for both samples. The electron density of CBM301 and VdP301 is $3.6 \times 10^{11}$ cm$^{-2}$, and for CBM302 and VdP302 is $1.7 \times 10^{11}$ cm$^{-2}$, as determined by magneto-transport measurements of Shubnikov de-Haas (SdH) oscillations at low magnetic fields, see SM. Their mobilities are $27.8 \times 10^6$ cm$^2$(Vs)$^{-1}$ and $20.3 \times 10^6$ cm$^2$(Vs)$^{-1}$, respectively.

The electrical contacts for the Corbino samples were patterned using UV lithography followed by e-beam deposition of Ge/Au/Ni/Au layers of 26/54/14/100 nm thickness and at a $1-2$ A s$^{-1}$ rate. Subsequently, the contacts were annealed in H2N2 atmosphere using a two-step annealing procedure: a 20 s first step at 370 °C followed by a longer 80 s step at 440 °C[31,32].

Both Corbino samples have three 2DEG rings with inner/outer radii (150/750) μm, (960/1000) μm, and (1300/1400) μm, respectively, see Fig. 1a. One van der Pauw sample was cleaved from the exact same wafer used to fabricate CBM302, and from a twin wafer with an identical heterostructure grown on the same day for CBM301. They are $3 \times 3$ mm square wafers with eight diffused indium contacts around the perimeter. All samples were cooled to a base temperature of approximately 20 mK in a dilution refrigerator and were illuminated by a red LED from room temperature down to approximately 6 K to increase both the mobility and density of the 2DEG. Specific details pertaining to the circuit used to perform the transport measurements are provided in the SM.

## Data availability

The data presented in this work are available from the corresponding author upon reasonable request.

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

## Acknowledgements

This work has been supported by NSERC (Canada), FRQNT-funded strategic clusters INTRIQ (Québec), PromptInnov/MEI PSR Quantique (Québec) in partnership with Mitacs and Montreal-based CXC. The work at Princeton University is funded in part by the Gordon and Betty Moore Foundation's EPiQS Initiative, Grant GBMF9615 to L.N. Pfeiffer, and by the National Science Foundation MRSEC grant DMR 2011750 to Princeton University. Sample fabrication was carried out at the McGill Nanotools Microfabrication facility. We would like to thank B.A. Schmidt and K. Bennaceur for their technical expertise during the fabrication and the earlier characterization of the Corbino sample, and R. Talbot, R. Gagnon, and J. Smeros for general laboratory technical assistance.

## Author contributions

S.V. and G.G. conceived the experiment. K.W.W. and L.N.P. performed the semiconductor growth by molecular beam epitaxy and provided the material. S.V. fabricated the Corbino and prepared the VdP samples. S.V. performed the electronic transport measurement in the Corbino samples at low temperatures, with the assistance and expertise of F.P. and M.P. O.Y. assisted in noise reduction and pre-amplification. S.V. performed the data analysis, helped by Z.B.K. for the development of the computer routine. M.P.L. and T.S. provided important expertise regarding data acquisition, semiconductor expertise, and interpretation of the results. K.A. provided theoretical guidance. S.V. and G.G. wrote the manuscript, and all authors commented on it.

## Competing interests

The authors declare no competing interests.
