## [Peer Review File · Nature Communications]

REVIEWER COMMENTS

Reviewer #1 (Remarks to the Author):

The authors are reporting the possible experimental observation of the Gurzhi effect in high-mobility GaAs 2DEGs, a transition from ballistic transport to viscous flow characterized by an increase in resistance as temperature decreases.

The authors' result is significant to the field, and is timely following the recent predictions of Ahn and Das Sarma last year [Ref. 4 from main text]. The experimental methodology is mostly sound (except for item 2 below), and I broadly support publication in Nature Communications.

However, I would like the authors to address the following two issues:

(1) The authors do not explicitly exclude the possibility of percolation (or metal-insulator transition) being the physical origin for their observations. Below the transition temperature ($T < 0.5\text{K}$), all samples have a constant resistance (with respect to temperature), which rules out "insulator" behavior. This may be obvious to a specialist, but possibly not to the broad readership of Nature Communications. A single sentence in the main text would address this issue.

(2) The authors present a reasonable explanation for the behavior in samples (CBM/VdP) 302. However, an explanation for the behavior of sample CBM301 is still lacking. Can the authors have another go at finding an explanation?

I wonder if the following could explain the behavior of sample CBM301. The four-terminal setups A and B in the Corbino samples are not "perfect/ideal:" some voltage will be dropped/picked-up across the inner two circular ohmic contacts (whose metal has diffused all the way to the 2DEG). This is an issue encountered in transmission line measurements (TLM) across non-adjacent ohmic contacts. The four-terminal setup may thus be able to pick up the superconducting transition of AuGeNi used in GaAs ohmic contacts [1]. The resistance drop in Figure 4a in the manuscript looks eerily similar to Figure 3a from [1]. The scenario above could be tested by repeating measurements on sample CBM301 in the presence of an in-plane or out-of-plane magnetic field of 0.2 T, above the critical field of 0.15T for superconductivity in AuGeNi.

[1] "Cooling low-dimensional electron systems into the microkelvin regime;"

L. V. Levitin et al., Nature Communications 13, 667 (2022).

Reviewer #2 (Remarks to the Author):

The authors present a study on the electron transport of ultra-high mobility AlGaAs/GaAs 2DEGs at ultra-low temperatures in the presence of a magnetic field. The study presents some very interesting results with good methodologies, appropriate considerations of errors in experiments that are accounted for, and a good analysis of the experimental data. However, there are still a few points that need to be considered and addressed before the manuscript can be reconsidered for publication.

The comments for the authors, in no particular order, are:

- 1.) What is the heterostructure utilized in this study? What is the substrate, what are the buffer layers, is there a surface capping layer (if so, how thick?), and what doping is performed in these heterostructures (e.g., modulation doping)? These details are very helpful for the reader in interpreting the results and understanding how the electron channel and electrical contact is made and measured in these samples. A schematic of the heterostructure can be included in the supplementary section.
- 2.) It is recommended that corbino rings with smaller diameters (e.g., channel length) and one larger diameter be fabricated and the electrical transport properties measured. Given the very large mean free path and the current dimensions of the devices, this fabrication should be relatively straightforward. This would give more definitive insight into what the approximate mean free paths are in these heterostructures.
- 3.) The authors mention that a 40 um mean free path is not indicative of the bulk mean free path of electrons. This is a powerful statement. Do the authors have any notion for what the bulk mean free path should be then? There is likely not a classical analogue here as the authors mentioned, but more discussion on this aspect would improve the manuscript.

Response to the referee reports.

Reviewer #1 (Remarks to the Author):

The authors are reporting the possible experimental observation of the Gurzhi effect in high-mobility GaAs 2DEGs, a transition from ballistic transport to viscous flow characterized by an increase in resistance as temperature decreases.

The authors' result is significant to the field, and is timely following the recent predictions of Ahn and Das Sarma last year [Ref. 4 from main text]. The experimental methodology is mostly sound (except for item 2 below), and I broadly support publication in Nature Communications.

Authors' response: we thank the reviewer for his/her positive outlook on our work and for supporting publication in Nature communications.

However, I would like the authors to address the following two issues:

(1) The authors do not explicitly exclude the possibility of percolation (or metal-insulator transition) being the physical origin for their observations. Below the transition temperature ($T < 0.5\text{K}$), all samples have a constant resistance (with respect to temperature), which rules out "insulator" behavior. This may be obvious to a specialist, but possibly not to the broad readership of Nature Communications. A single sentence in the main text would address this issue.

Authors' response: we thank the reviewer for this comment. Indeed, it is known in the field that resistivity in ultra-high mobility 2DEGs (GaAs) saturates below 0.8K, or at lower temperatures in some cases. As suggested by the reviewer, we have added a sentence in the main text to address this.

(2) The authors present a reasonable explanation for the behavior in samples (CBM/VdP) 302. However, an explanation for the behavior of sample CBM301 is still lacking. Can the authors have another go at finding an explanation?

Authors' response: we understand this comment by the reviewer, and we would like to re-iterate that unfortunately we cannot clearly identify a single mechanism that would reconcile the anomalous behaviour seen in both samples. We have ruled out the "obvious":

- a) A potential electron density shift, by performing SdH;
- b) Non-equilibrium, or thermal: we have verified that indeed the "signal" is at equilibrium by performing discrete magnetic field measurements.
- c) Concerns about residual field trapped in the superconducting magnet were investigated and ruled out.

Thanks to the work by Ahn and Sarma, the Knudsen parameter is roughly falls below 0.5 in CBM302 at 550 mK (where the anomalous transport is observed) while in CBM301 it falls below 0.5 at 850 mK, and it is interesting to note that after the "hump" in CBM301 there is slight decrease in resistivity followed by a rather large unusual resistivity saturation (in temperature). These latter two are consistent with hydrodynamics playing some role, although we have not yet been able to identify the cause for the sudden resistance increase. We also note that both samples were made in the exact same way, they have extremely well defined ohmic resistances (2-6 Ohms), and their magneto-resistances in the FQH regime is exemplary. For instance, CBM301 showed spectacular "bubble" and "stripe" phases, as well as competing phases in the flank of the 5/2 FQH, and these were published by us [Phys. Rev. Lett. 120, 136801 (2018)]. We also provide below the magnetoresistance in the 2nd Landau level for CBM302 (to be published).

A tentative explanation we have explored is sample electron density fluctuation [*Nano Letters* 19 (3), 1908-1913 (2019)] (and mobility electron mobility fluctuations) over some length scale which could perhaps lead to an interference phenomenon at the cross-over between super-ballistic to hydrodynamic transport. Interestingly, in wider 2-point Corbino samples, entropy deduced from time-resolved specific heat experiments agree quantitatively with thermopower measurements in large hall bars from the Caltech group [see our previous work *Phys. Rev. B* 95, 201306(R) (2017)]. However, given the channel width in the CBM samples being significantly smaller (40 microns rather than 750 microns in the 2-point), we cannot rule out there could be an “easier current path” and, perhaps, a resonance or interference. While this is highly speculative, we have added a small discussion to address this in the main text.

I wonder if the following could explain the behavior of sample CBM301. The four-terminal setups A and B in the Corbino samples are not “perfect/ideal:” some voltage will be dropped/picked-up across the inner two circular ohmic contacts (whose metal has diffused all the way to the 2DEG). This is an issue encountered in transmission line measurements (TLM) across non-adjacent ohmic contacts. The four-terminal setup may thus be able to pick up the superconducting transition of AuGeNi used in GaAs ohmic contacts [1]. The resistance drop in Figure 4a in the manuscript looks similar to Figure 3a from [1]. The scenario above could be tested by repeating measurements on sample CBM301 in the presence of an in-plane or out-of-plane magnetic field of 0.2 T, above the critical field of 0.15T for superconductivity in AuGeNi.

[1] “Cooling low-dimensional electron systems into the microkelvin regime;”
L. V. Levitin et al., *Nature Communications* 13, 667 (2022).

Authors’ response: we thank the referee for pointing this out, and we became aware of potential superconductivity in NiGeAu recently (*Nature Communications* 13, 667 (2022) and *Appl. Phys. Lett.* 117, 162104 (2020)). While the presence of a supercurrent could explain potentially the “hump” observed in CBM301, it certainly would not explain the behaviour observed in CBM302. Nevertheless, to rule it out, we have:

- a) Carefully measured the conductance in a perpendicular field in both Corbino near the expected critical field (as suggested by the referee) and found no evidence of a critical field whatsoever. These data have been added in the supplementary information material, and a sentence has been added in the main text, and references added.
- b) For CBM301, we have measured both two-point and four-point differential resistance with a DC bias (dV/dI versus I_{dc}) and found no evidence of any non-linearity in the I-V (at zero magnetic field). Rather, ohmic behaviour was observed at the base temperature of our Bluefors dilution refrigerator.

While we certainly believe the work by the authors above is correct, we think it is most likely highly dependent on the “recipe” used to form the NiGeAu ohmic contacts. For instance, neither us, nor our collaborators at Sandia Laboratories, have observed superconductivity in contacts even though several of our past (and current) measurements are inherently two points [see for example *Science* **343**, 631 (2014)]. Comparing with the recipe from the London/Cavendish groups, our NiGeAu recipe differs significantly, which may explain why we do not observe superconductivity. Although the recipe is included in some references we cited, we provide it explicitly below.

Ge/Au/Ni/Au in 26/54/14/100 nm layers at a rate of 1-2 Å/s.
 Annealing 440°C for 80 sec

Annealing details (step by step):

- Ramp till 370°C at 10°C/sec.
- Stay at 370°C for 20 sec.
- Use ramping rate 10°C /sec for ramping from 370°C to 440°C
- Stay at 440°C for 80 sec
- 20 sec for ramping down.

We thank the reviewer for pointing this out, and we have added a small discussion in the main text to address this.

Finally, we would like to thank the Reviewer once again for the thorough review which has led to an improved manuscript which we hope can now obtain final approval for publication in *Nat. Comm.*

Reviewer #2 (Remarks to the Author):

The authors present a study on the electron transport of ultra-high mobility AlGaAs/GaAs 2DEGs at ultra-low temperatures in the presence of a magnetic field. The study presents some very interesting results with good methodologies, appropriate considerations of errors in experiments that are accounted for, and a good analysis of the experimental data. However, there are still a few points that need to be considered and addressed before the manuscript can be reconsidered for publication.

Authors' response: we thank the reviewer for his/her positive outlook on our work and for supporting publication in *Nature communications*.

The comments for the authors, in no particular order, are:

1.) What is the heterostructure utilized in this study? What is the substrate, what are the buffer layers, is there a surface capping layer (if so, how thick?), and what doping is performed in these heterostructures (e.g., modulation doping)? These details are very helpful for the reader in interpreting the results and understanding how the electron channel and electrical contact is made and measured in these samples. A schematic of the heterostructure can be included in the supplementary section.

Authors' response: this is an excellent point, and we apologize if we omitted to add the complete heterostructure in the original version. In the past, we have carefully studied the relationship between mobility, density, and "growth details" such as the setback distance [see Appl. Phys. Lett. **96**, 162112 (2010)] and we wholeheartedly agree that details are of paramount importance here. A complete sketch of the heterostructure is now provided in the SM for both samples.

In hindsight, we would like to summarize the main differences between CBM301 and CBM302:

- a) The electron density ($3.6 \times 10^{11} \text{ cm}^{-2}$ for CBM301 and $1.7 \times 10^{11} \text{ cm}^{-2}$ for CBM302) leading to a distinctive electro-electron interaction parameter (r_s);
- b) The quantum well width (30 nm for CBM301 and 40 nm for CBM302);
- c) The setback distance for the dopants (80.54 nm for CBM301 and 162.54 nm for CBM302).

We have also added a sentence in the main text clarifying the main differences between both samples.

Finally, we would like to add that both samples show exemplary behaviour in a magnetic field. CBM301 showed spectacular "bubble" and "stripe" phases as well as competing phases in the flank of the 5/2 FQH, and these were published by us [Phys. Rev. Lett. **120**, 136801 (2018)]. We also provided above in our response to Reviewer #1 the magneto-resistance in the 2nd Landau level for CBM302 (to be published).

2.) It is recommended that corbino rings with smaller diameters (e.g., channel length) and one larger diameter be fabricated and the electrical transport properties measured. Given the very large mean free path and the current dimensions of the devices, this fabrication should be relatively straightforward. This would give more definitive insight into what the approximate mean free paths are in these heterostructures.

We fully agree with this suggestion, and this will be the subject of future work. Nailing down "what is the mean free path" is extremely important especially given the recent advance by Dr. Pfeiffer who can now grow 2DEGs with mobility as high as $57 \times 10^6 \text{ cm}^2/(\text{V s})$. But as I am sure the reviewer knows, these 2DEGs are not uniform, and measurements performed by Dr. Pfeiffer show clear spatial fluctuations [*Nano Letters* **19** (3), 1908-1913 (2019)].

We would like to add that we first observed anomalous temperature dependence in a two-point Corbino with a relatively large channel width of 750 micron. But in this case, the measurements being inherently two-point, we were not able to rule out a contribution from the Ohmic contacts, which led us to investigate the four-point CBM.

We thank the reviewer and we have added a small discussion in the main text regarding the importance of determining "what is the mean free path" in these ultra-high mobility 2DEGs (same discussion as in the light of the point 3).

3.) The authors mention that a 40 um mean free path is not indicative of the bulk mean free path of electrons. This is a powerful statement. Do the authors have any notion for what the bulk mean free path should be then? There is likely not a classical analogue here as the authors mentioned, but more discussion on this aspect would improve the manuscript.

We thank the reviewer for making this comment, as we believe the manuscript was not sufficiently clear on this topic. What we meant to say is that the momentum relaxing mean free path, when neglecting phonons at temperatures less than 0.8K or so, is exceedingly larger than the channel width, i.e. 271 microns for CBM301 and 138 microns for CBM302. On top of that, the momentum conserving mean path l_{ee} falls below 40 microns at ~550 mK for CBM302 according to Anh and Das Sarma. Thus, both mean free paths are either larger, or close to a 40 microns channel in the temperature range between 20 mK and ~1K. While the ~3 mm scale Van der Pauw showed the expected electronic transport behavior in that temperature range, the opposite cannot be truer in the 40 microns Corbino.

The situation is rendered even more complex in the light of very recent work (and years of internal knowledge) by Dr. Pfeiffer on sample inhomogeneity over relatively smaller scales, e.g ~100 microns as quoted in Ref. [*Nano Letters* **19** (3), 1908-1913 (2019)] in a high mobility 2DEGs. As the sample quality increases, as defined

by mobility at $B=0$ BUT also magneto-transport and observation of better formed, or new fragile many-body states, the understanding of what is the dominant scattering mechanism is arguably extremely important.

We have added clarity to the paper, both in the section on transport lengthscales, as well as in the conclusion and thank the reviewer for making the comment.

Finally, we would like to thank the reviewer once again for the thorough review which has led to an improved manuscript which we hope can now receive final approval.

List of changes implemented.

A red text colour is used in the manuscript to indicate the changes in the main material.

Main text

- 1) A new sentence added in section 'Experimental design and procedure'.

Added sentence: A sketch of the heterostructure that include the main growth parameters such as the setback distances for the dopants and the capping layer thickness is provided in the supplementary material (SM) for both samples.

- 2) Figure 4 modification: In Figure 4, a guide to eye dotted line is added. And added a sentence in the caption.

Added sentence in the caption: A guide-to-the-eye blue dash line shows the moving average of the resistance at low temperature for CBM301 and CBM302.

- 3) New sentences are added to 'Comparison of Corbino and van der Pauw measurements.' Section

Added sentence: We also note that in all cases the resistance is nearly constant at temperatures below ~ 0.5 K, as expected in very high-mobility GaAs/AlGaAs 2DEGs, hence ruling out the anomalous behaviour in electronic transport observed being caused by a percolation process (metal-insulator transition). Recent work also found that under some conditions NiGeAu could be superconducting with a transition temperature around ~ 700 mK and a critical field of ~ 0.15 T [8, 9]. In order to investigate if a supercurrent and/or a proximity effect could have played a role in our transport measurements, two-terminal and four-terminal differential resistance measurements under a DC current bias were performed at base temperature of the dilution refrigerator ($T \sim 20$ mK) and the $I - V$ obtained were found to be linear (ohmic). In addition, we have performed magneto-transport measurements in a perpendicular field near the expected critical field 0.15 T (also at base temperature) and found no evidence of superconductivity in either samples, see SM. Finally, additional electronic transport data up to 100 K in the Corbino and up to 10 K in the VdP samples are provided in the SM that illustrates the distinctive behaviour between the VdP and CBM samples at temperatures below a few Kelvins.

- 4) A modified sentence added in section Gurzhi effect.

Added sentence: More recent works have focused on bilayers [15], strongly-correlated 2D hole systems [16], non-local measurements [17, 18] including an extensive study of the conductivity in GaAs/AlGaAs channel with smooth sidewalls and perfect slip boundary condition [19], graphene [20-23], graphene Corbino rings [24] as well as theoretical interest of hydrodynamic in Corbino geometries [25-28].

- 5) New sentences added before conclusion.

Added sentence: A recent study conducted on spatial mapping of local electron density fluctuation in a high-mobility GaAs/AlGaAs 2DEG by way of scanning photoluminescence [29] reported electron density variations

up to 100 μm with a spot size of 40 μm . These fluctuations are likely to generate local electron mobility fluctuations, and we hypothesize that it could perhaps play a role in a Corbino measurement scheme because there is no edge and the concentric sample can be viewed as a very large number of conductors wired in parallel. This being said, whether a higher conductance path due to a local spatial fluctuation in the electron density (or mobility) would occur, and lead to the anomalous electronic transport observed in both CBM301 and CBM302 remains an open question. This will be the subject of future works.

- 6) Modified sentences added in conclusion.

Modified sentence: **this work demonstrates that a 40 μm channel length is not bulk in very high mobility 2DEGs since both the momentum conserving and momentum relaxing mean free path values are either larger, or equal to the channel length in the 20 mK – ~1 K temperature range.**

- 7) Included data availability statement with a title of '**DATA AVAILABILITY**'

Reference section

- 1) Added new references for clarity and completeness.

Reference numbers: **8, 9, 25, 26, 27, 28 and 29**

Supplementary material

- 1) Added two new sections and two new figures.

Sections added: **Heterostructure** and **Magneto-transport measurement**.
Figures added: **S1** and **S4**

- 2) Modified figure S3.

REVIEWERS' COMMENTS

Reviewer #1 (Remarks to the Author):

Accept for publication in Nature Communications.

Reviewer #2 (Remarks to the Author):

The manuscript has been improved and the comments have been addressed. The addition of the prior reported spatial variation of electron density is an important point.

Response to the referee reports.

REVIEWERS' COMMENTS

Reviewer #1 (Remarks to the Author):

Accept for publication in Nature Communications.

Response: We thank Reviewer #1 for accepting the manuscript for publication. We would also like to thank the reviewer for the through review and valuable comments which helped to improve the manuscript.

Reviewer #2 (Remarks to the Author):

The manuscript has been improved and the comments have been addressed. The addition of the prior reported spatial variation of electron density is an important point.

Response: We thank Reviewer #2 for accepting the manuscript for publication. We would also like to thank the reviewer for the through review and valuable comments which helped to improve the manuscript.